# Benzalkonium Chloride Induces a VBNC State in *Listeria monocytogenes*

**DOI:** 10.3390/microorganisms8020184

**Published:** 2020-01-28

**Authors:** Matthias Noll, Katharina Trunzer, Antje Vondran, Szilvia Vincze, Ralf Dieckmann, Sascha Al Dahouk, Carolin Gold

**Affiliations:** 1Institute for Bioanalysis, Department of Applied Sciences, Coburg University of Applied Sciences and Arts, Friedrich-Streib-Straße 2, D-96450 Coburg, Germany; Katharina.trunzer@hs-coburg.de (K.T.); Antje.vondran@hs-coburg.de (A.V.); gold.car@t-online.de (C.G.); 2German Federal Institute for Risk Assessment, Max-Dohrn-Str. 8-10, 10589 Berlin, Germany; Szilvia.Vincze@bfr.bund.de (S.V.); Ralf.Dieckmann@bfr.bund.de (R.D.); Sascha.Al-Dahouk@gmx.de (S.A.D.)

**Keywords:** 2-NBDG, antibiotic susceptibility, benzalkonium chloride, colony forming units, flow cytometry, *Listeria monocytogenes*, metabolic activity, viable but nonculturable, VBNC

## Abstract

The objective of our study was to investigate the effects of benzalkonium chloride (BC) adaptation of *L. monocytogenes* on the susceptibility to antimicrobial agents and on the viable but non culturable (VBNC) state of the bacterial cells. We adapted *L. monocytogenes* SLCC2540 to BC by applying BC below minimum inhibitory concentration (MIC) to above minimum bactericidal concentration (MBC). The culturable fractions and the susceptibility of adapted and parental cells to BC were assessed. In addition, cell membrane permeability and glucose uptake were analyzed by multi parametric flow cytometry using the fluorescent agents SYTO9, propidium iodide, and 2-deoxy-2-[(7-nitro-2,1,3-benzoxadiazol-4-yl)amino]-D-glucose (2-NBDG). Adapted cells displayed a two-fold MIC increase of BC and reduced antibiotic susceptibility. At high BC concentrations, the decrease in the number of colony forming units was significantly lower in the population of adapted cells compared to parental cells. At the same time, the number of metabolically active cells with intact membranes was significantly higher than the number of culturable cells. Growth-independent viability assays revealed an adapted subpopulation after BC application that was not culturable, indicating increased abundance of viable but nonculturable (VBNC) cells. Moreover, adapted cells can outcompete non-adapted cells under sublethal concentrations of disinfectants, which may lead to novel public health risks.

## 1. Introduction

*Listeria monocytogenes* is a ubiquitous, Gram-positive, facultative intracellular opportunistic pathogen and the causative agent of human listeriosis, a disease with a wide variety of clinical presentations ranging from mild fever to meningoencephalitis with lethal outcome. Populations at highest risk for invasive listeriosis include elderly and immunocompromised persons, and pregnant women and their newborns [1]. The incubation period of listeriosis may be long (median 11 days, range 0–70 days) [1]. *Listeria monocytogenes* is usually transmitted by the consumption of contaminated food [2,3], and various foodstuffs have been identified as vehicles [2,4,5,6]. Adverse conditions may induce tolerance or resistance towards environmental stress factors and/or a shift in *L. monocytogenes* from a culturable state to a viable but nonculturable state (VBNC) [7]. VBNC cells are defined as bacterial cells that do not form colonies on standard culture media but still retain metabolic activity [8] and may revert to the active state [9]. Early VBNC studies coupled bacterial culture with microscopic counting of stained cells using SYTO 9/propidium iodide (SYTO9/PI), 5-cyano-2,3-ditolyl tetrazolium chloride (CTC), 4′,6-diamidino-2-phenylindole (DAPI), and/or fluorescently labelled antibodies [7,10,11,12]. Recent studies expanded the methodological approaches to determine the metabolic activity and cell membrane integrity of VBNC cells by ATP determination [13], quantitative PCR combined with propidium monoazide treatment [14], the capability to ferment sugars after reuptake [13], and flow cytometric analyses coupled with cell staining [15]. Flow cytometric analysis of SYTO9/PI was also linked to the uptake activity of 2-deoxy-2-[(7-nitro-2,1,3-benzoxadiazol-4-yl)amino]-D-glucose (2-NBDG) to assess both cell membrane integrity and metabolic activity simultaneously [16], thereby quantifying VBNC bacteria at the single cell level [17]. Various stress conditions have been identified to induce the VBNC state in *L. monocytogenes*, including chlorine-based sanitizing treatments [15] and non-ionic surfactants combined with inorganic salts [13].

Quaternary ammonium compounds (QACs), such as benzalkonium chloride (BC), are widely used as biocides for the disinfection of surfaces, including food production environments [18,19]. The alkyl chain of QACs perturbs and disrupts the microbial membrane bilayer and its charge distribution [20]. Microbial mechanisms of tolerance and resistance towards QACs include changes in the overall membrane composition, downregulation of porins, overexpression or modification of efflux pumps, horizontal gene transfer of resistance-associated genes, biofilm formation, and biodegradation of QACs [18]. Exposure of *L. monocytogenes* to progressively increasing BC concentrations frequently leads to BC adaptation, which can often be attributed to shifts in the efflux pump activity [19,21,22]. However, how BC adaptation affects the VBNC state and the metabolic activity of *L. monocytogenes* has not been addressed so far. 

Therefore, the aims of our study were to investigate the effects of BC adaptation in *L. monocytogenes* on (i) susceptibility to antimicrobial agents (BC and antibiotics), (ii) culturability, and (iii) metabolic activity. 

## 2. Materials and Methods 

### 2.1. Listeria monocytogenes Isolate under Study and Biocide Susceptibility Testing

*Listeria monocytogenes* SLCC2540 was investigated in this study. It was isolated from human in the USA in 1956 and belonged to serotype 3b. Minimum inhibitory (MIC) and minimum bactericidal concentrations (MBC) were determined in nine independent replicates, as previously outlined [23] with minor modifications. Brain heart infusion broth (BHI) (Carl Roth GmbH, Karlsruhe, Germany) was utilized, and the cell density of the inoculum was adjusted to approx. 2 × 10^8^ cells mL^−1^. Dey-Engley neutralizing broth (Sigma-Aldrich KGaA, Darmstadt, Germany) was used to quench biocidal effects for MBC testing [23]. A stock solution of 4 mg mL^−1^ BC (Sigma Aldrich) was prepared freshly by dissolving 0.02 g of BC in 5 mL of sterile distilled water. 

### 2.2. Adaptation to Benzalkonium Chloride

For bacterial adaptation, *L. monocytogenes* SLCC2540 was cultivated by serial passages in 200 µL BHI in the presence of increasing BC concentrations starting from 2 µg mL^−1^. After each incubation cycle of 24 h, the optical density (OD) was measured at a wavelength of 595 nm after 5 s of shaking using a FLUOstar OMEGA microplate reader (BMG Labtech, Ortenberg, Germany). The empty value (microtiter well containing biocide solution and BHI without *L. monocytogenes*) was subtracted from measured data, and a ΔOD_595 nm_ of 0.1 was considered as the cutoff value for bacterial growth, if growth control (microtiter without biocide in solution and BHI with *L. monocytogenes*) had a ΔOD_595 nm_ > 0.4. When bacterial growth was observed, strain SLCC2540 was subcultured in identical BC concentrations or stepwise (by 1 μg mL^−1^) increasing BC concentrations until growth inhibition was achieved. The stability of biocide tolerance was tested after 15 subcultures of the adapted strain. Parent and adapted strains were stored in BHI broth supplemented with 15% glycerol (Carl Roth) at −80 °C.

### 2.3. Determination of Culturable Cells after BC Exposure

Parent and adapted *L. monocytogenes* strains (inoculum: 2 × 10^8^ cells mL^−1^) were incubated and gently shaken in 200 µL BHI without BC and in BHI supplemented with BC concentrations of 2, 3, 4, 5, 6, 8, 9, 10, 11, 12, and 13 μg mL^−1^ at 37 °C for 24 h. Colony forming units (CFUs) were determined in nine replicates according to the standard protocol of ISO 11290 [24]. 

### 2.4. Antibiotic Susceptibility Testing and Flow Cytometric Analysis

Parent and adapted *L. monocytogenes* SLCC2540 cells were incubated in a range of BC concentrations (0, 2, 3, 4, 5, 6, 8, 9, 10, 11, 12, and 13 μg mL^−1^) at 37 °C for 24 h. All experiments were conducted in biologically independent triplicates. A total of 1.5 mL of each liquid culture was centrifuged at 10,000× *g* at 4 °C for 5 min. Afterwards, cells were washed three times with sterile filtered phosphate buffered saline (PBS; 8 NaCl gL^−1^, 0.2 KCl gL^−1^, 1.44 Na_2_HPO_4_ gL^−1^, 0.24 KH_2_HPO_4_ gL^−1^; pH 7.4) and finally resuspended in 1.5 mL PBS. Each sample was divided into four aliquots. The first fraction was used to test the antibiotic susceptibility towards 14 different antibiotics, as previously described [25], using BHI broth instead of H-medium. The remaining three fractions were analyzed by flow cytometry. One aliquot was stained with PI (LIVE/DEAD BacLight kit for flow cytometry, Fisher Scientific, Schwerte, Germany), another one with SYTO9 (Life Technologies GmbH, Darmstadt, Germany), and the last one with the fluorescence labelled glucose analogue 2-NBDG (Life Technologies) modified according to Berney et al. [16]. Briefly, flow cytometry was performed using a NovoCyte Flow Cytometer (Acea Biosciences Inc., San Diego, CA, USA) at an excitation wavelength of 488 nm. A photomultiplier with a band pass filter of 530/20 nm was used to collect the green fluorescence of SYTO9 and 2-NBDG, while the red fluorescence of PI was detected using a band pass filter of 615/20 nm. Preliminary studies on *L. monocytogenes* strain SLCC2540, not exposed to BC, revealed linear uptake kinetics when < 5 × 10^9^ cells mL^−1^ were incubated with 10 μM 2-NBDG for 15 min at 37 °C. Data obtained from each channel were displayed in logarithmic scale and analyzed using the Novo Express software 1.2 (Acea Biosciences Inc.). For each sample, we collected parameters of 300,000 events at a flow rate of 3000 events s^−1^. The respective volume was documented to receive the amount of events per mL, namely flow cytometric cell counts. 

*L. monocytogenes* strain SLCC2540 incubated in BHI to logarithmic growth phase served as viable positive control, while cells heated at 95 °C for 10 min were used as non-viable negative control. Cell densities of both controls were set to 2 × 10^8^ cells mL^−1^ in PBS. Controls were used to evaluate the uptake kinetics of PI, SYTO9, and 2-NBDG and for adjusting quadrants. Quadrants were associated with physiological properties such as cell membrane integrity (intact: SYTO9 positive but PI negative; not intact: SYTO 9 positive and PI positive) and metabolic activity (active: 2-NBDG positive; inactive: 2-NBDG negative). Glucose uptake rates of parent and adapted *L. monocytogenes* cells were calculated by dividing fluorescence intensity of 2-NBDG by total cell count in each sample. Mean values and standard deviations are presented.

### 2.5. Statistics 

Statistical analyses were performed to evaluate significant differences between adapted and parent *L. monocytogenes* SLCC2540 cells as well as between CFUs and cytometric counts of stained cells using R software [26]. Two way-analysis of variance (ANOVA) and Tukey’s post-hoc test were performed using Origin 2017 (OriginLab Corporation, Northampton, MA, USA). The significance level was set to *p* ≤ 0.05. 

## 3. Results

### 3.1. Adapted L. monocytogenes Cells Show Shifts in MICs of BC and Some Antibiotics 

In pretests, the neutralizer used for MBC determination proved to be effective for BC and was not toxic to *L. monocytogenes* (data not shown). Susceptibility testing of *L. monocytogenes* SLCC2540 revealed a MIC of 4 µg mL^−1^ and a MBC of 11 µg mL^−1^. After adaptation to BC, the adapted cells exhibited a MIC of 8 µg mL^−1^, while the MBC was still the same. The susceptibility profile of the adapted cells remained stable following fifteen subcultures. 

CFUs were determined to investigate the fraction of culturable cells. For the original strain SLCC2540 (parent cells), CFUs increased compared to the starting inoculum during growth in BHI without BC and with BC below MIC concentrations (Figure 1A). Incubation of SLCC2540 in BHI with BC concentrations above MIC resulted in a reduced number of CFUs. For adapted cells, the CFU counts showed a similar pattern as for the parent cells during incubation in sub-inhibitory BC concentrations. Nonetheless, the CFUs of adapted cells were higher compared to parent cells after incubation in BHI, with BC concentrations ranging between 6 and 9 μg mL^−1^.

Total flow cytometric cell counts of SYTO9-positive cells were higher at sub-inhibitory concentrations compared to the primary inoculum for both cell types (Figure 1B). At concentrations equal to or higher than the respective MIC of BC, the total number of cells resembled the initial inoculum.

Adapted cells showed a reduced susceptibility to certain antibiotics, mainly ceftriaxone, gentamicin, linezolid, tetracycline, and trimethoprim/sulfamethoxazole, compared to parent cells (Table 1). Moreover, the increase in antibiotic susceptibility with increasing BC concentrations was less pronounced in adapted cells compared to parent cells.

### 3.2. Adaptation Increased the Proportion of Cells with Intact Membranes at high BC Concentrations

The proportion of cells with intact membranes was determined by calculating the ratio of SYTO9-positive and PI-negative cells [16,17]. Between 80% and 100% of parent and adapted cells were intact after cultivation in BHI without and supplemented with up to 6 μg mL^−1^ BC (Figure 2, Appendix A). At higher BC concentrations (8 to 11 μg mL^−1^), the proportion of intact cells was significantly higher for adapted cells compared to parent cells.

### 3.3. Adapted Cells Maintain Metabolic Activity at High BC Concentrations

To quantify metabolic activity of *L. monocytogenes* after biocide exposure, we analyzed 2-NBDG uptake rates of parent as well as adapted cells following cultivation in BHI supplemented with BC. 2-NBDG uptake was similar in parent and adapted cells when incubated in BC concentrations below the respective MICs. While the number of metabolically active cells decreased for the parent strain at BC concentrations > 4 μg mL^−1^, their number remained high in the adapted strain up to 11 μg mL^−1^ BC (Figure 3, Appendix A).

## 4. Discussion

Following adaptation to BC, *L. monocytogenes* SLCC2540 revealed a twofold MIC increase of this biocide compared to parent cells. In previous studies, similar and sometimes larger MIC increases after BC exposure were observed [22,27,28].

In addition, reduced susceptibility to ceftriaxone, gentamicin, linezolid, tetracycline, and the combination of trimethoprim and sulfamethoxazole emerged in BC-adapted cells. The development of reduced antibiotic susceptibility is in line with previous studies showing ciprofloxacin resistance in *L. monocytogenes* after BC exposure [29]. Rakic-Martinez et al. (2011) found that both the adaptation to ciprofloxacin and BC may induce mutations in efflux pump systems, which are responsible for multidrug resistances [29]. Susceptibility to antibiotics increased after incubation with increasing BC concentrations (Table 1), suggesting that pre-incubation with BC weakened the bacterial cells.

Phenotypic characterization of *L. monocytogenes* field isolates from German food production plants showed no cross-resistance between BC and antibiotics [23], indicating that regular disinfection measures do not necessarily induce cross-adaptation events. The BC-tolerant phenotype remained stable after multiple sub-cultures in drug-free medium, which has been shown for other *L. monocytogenes* strains [27,28]. This phenomenon suggests that phenotypic changes are most likely based on a genetically determined tolerance mechanism rather than on a short-term phenotypic adaptation. Several mechanisms of BC resistance have been recently reviewed [18]. However, the effect of BC adaptation on the development of a VBNC state has not been addressed in *L. monocytogenes* so far. By using flow cytometry, we were able to show that BC-adapted cells maintained their metabolic activity and intact cell walls in higher numbers than parent cells during cultivation in BHI supplemented with BC concentrations above the MIC of the parent strain. However, only a relatively low number of cells was culturable, indicating that increasing BC concentrations induced adapted cells to enter the VBNC state.

Flow cytometric analysis has the advantage of allowing markable phenotypic attributes to be visualized at the single cell level with a high precision and sensitivity. We did not detect an effect of BC on the fluorescence properties of PI, SYTO9, or 2-NBDG up to a concentration of 13 μg mL^−1^ (data not shown). The quadrants for the different phenotypic attributes were set by measuring the same cells without BC stress from the exponential growth phase (proliferating living cells with intact cell membrane integrity and metabolic 2-NBDG uptake activity) before and after heat treatment (dead cells with damaged cell membrane and no 2-NBDG uptake), thereby enabling a differentiation between metabolically active and intact cells from other cells and their quantification. Metabolic activity based on 2-NBDG uptake is commonly presented as a proportion in relation to external standards of dead and living cells [16,17]. In our study, controls were directly taken from the exponential growth phase of bacterial cells and therefore the external standard of living cells represented an ideal metabolic state. In contrast, a few cells of our external standard of heat-inactivated dead cells were used as an external standard of dead cells. Quantification of 2-NBDG-positive cells was based on a linear increase of the fluorescence signal as evaluated by living and dead *L. monocytogenes* cells without contact to BC. The pitfalls and benefits of flow cytometric analysis of bacterial cells after antimicrobial treatments were previously reviewed by Léonard and colleagues [30].

The VBNC state was first described by Xu et al. (1982) for *Escherichia coli* and *Vibrio cholerae* [12]. Since then, more than 85 other bacterial species were found to enter the VBNC state [31]. The VBNC state has been reported for a broad phylogenetic diversity, indicating a common prokaryotic mechanism to withstand environmental stress conditions. Chemical stress by sub-optimal pH, ethanol, chlorine, household cleaners, antibiotics, or biocides is able to induce the VBNC state in many bacterial species. Recently, the induction of a VBNC state in *L. monocytogenes* by household cleaners was analyzed [13]. *L. monocytogenes* cells almost instantly entered the VBNC state following exposure to a combination of non-ionic surfactants and inorganic salts and thereafter could not be resuscitated [13]. The vast majority of adapted cells in our study were viable and metabolically active but not detectable by standard cultivation techniques. It is alarming that *L. monocytogenes* cells in VBNC state induced by stress factors comparable to BC remain infectious [32], indicating an unseen (by cultivation techniques) but potent threat in industrial, clinical, and domestic environments. 

## 5. Conclusions 

BC induced the VBNC state and reduced antibiotic and BC susceptibility in adapted *L. monocytogenes* SLCC2540 cells. Selective environmental conditions may facilitate additional adaptations leading to multi-resistant strains. Moreover, adapted cells can outcompete non-adapted cells under sublethal concentrations of disinfectants, which may lead to novel public health risks. Therefore, the prevalence of adaptations to biocides and accompanying shifts in phenotypic features should be monitored over time, especially in high-risk environments, such as food processing plants and hospitals. These surveillance data have to be considered in the indispensable future definitions of biocide breakpoints. 

## Figures and Tables

**Figure 1 microorganisms-08-00184-f001:**
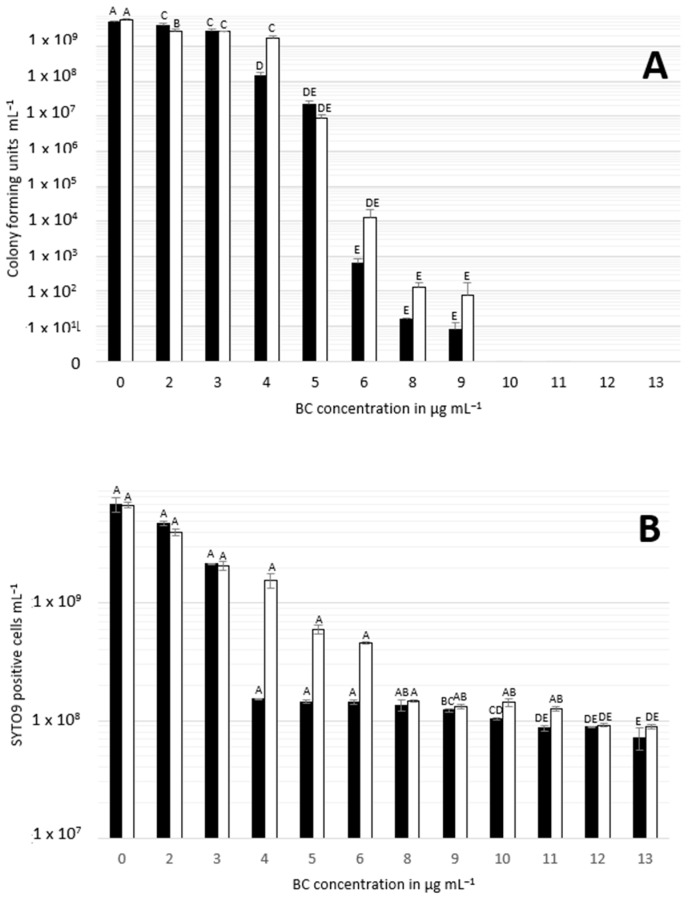
Colony forming units (**A**) and SYTO9-positive cells (**B**) of parent (black bars) and adapted (white bars) *Listeria monocytogenes* SLCC2540 following cultivation in brain heart infusion broth (BHI) containing benzalkonium chloride (BC) in various concentrations. Error bars indicate standard deviation of nine replicates. Different letters above bars within panels indicate significant differences (*p* < 0.05) according to two-way analysis of variance (ANOVA).

**Figure 2 microorganisms-08-00184-f002:**
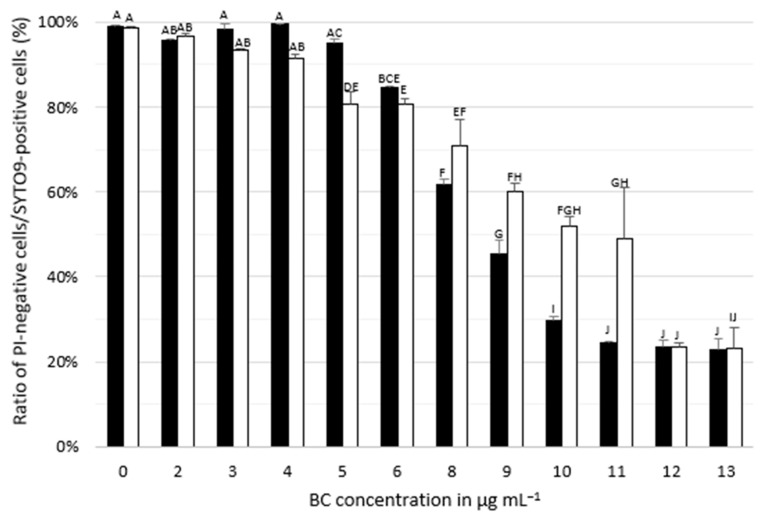
Mean ratio of PI-negative to SYTO9-positive parent (black bars) and adapted (white bars) *Listeria monocytogenes* SLCC2540 cells following cultivation in BHI containing benzalkonium chloride (BC) in various concentrations. SYTO9-positive cells are indicative of total cell counts, while PI-negative cells are a surrogate for intact cell membranes. Error bars indicate the standard deviation of three replicates. Different letters above bars within panels represent significant differences (*p* < 0.05) according to two-way analysis of variance (ANOVA).

**Figure 3 microorganisms-08-00184-f003:**
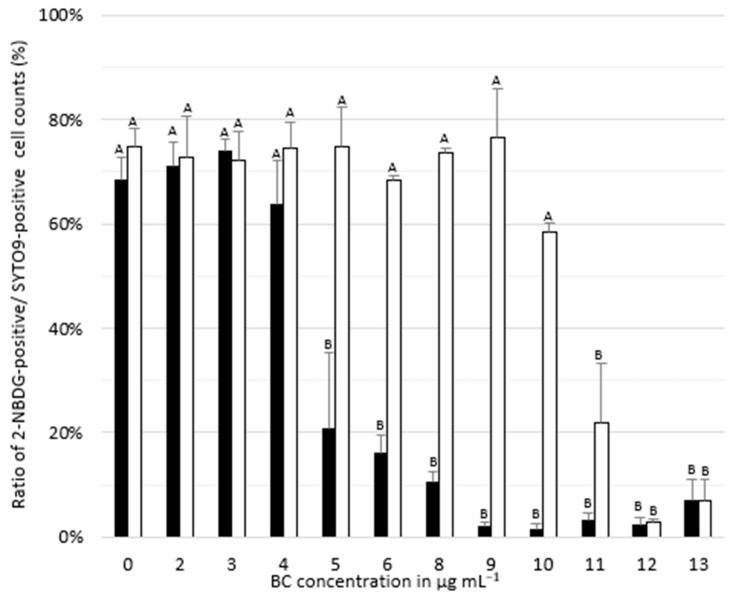
Mean ratio of 2-NBDG-positive to SYTO9-positive parent (black bars) and adapted *Listeria monocytogenes* SLCC2540 cells (white bars) following cultivation in brain heart infusion broth containing benzalkonium chloride (BC) in various concentrations. 2-NBDG-positive cells are indicative of metabolically active cells, while SYTO9-positive cells represent the total number of cells. Error bars show the standard deviation of three replicates. Different letters above bars within panels indicate significant differences (*p* < 0.05) according to two-way analysis of variance (ANOVA).

**Table 1 microorganisms-08-00184-t001:** Antibiotic susceptibility profiles of adapted (**A**) and parent (**B**) *Listeria monocytogenes* SLCC2540 cells after growth in BHI supplemented with benzalkonium chloride (BC). Differences in the susceptibility profiles of adapted and parent cells are highlighted in bold. Antibiotics are AMP, ampicillin; PEN, benzylpenicillin; CRO, ceftriaxone; CIP, ciprofloxacin; DPT, daptomycin; ERY, erythromycin; GEN, gentamicin; LIZ, linezolid; MER, meropenem; RIF, rifampicin; TET, tetracycline; TGC, tigecycline; T/S, trimethoprim/sulfamethoxazole; VAN, vancomycin.

**Minimum Inhibitory Concentrations in µg mL^−1^**													**A**	
**VAN**	**T/S**	**TGC**	**TET**	**RIF**	**PEN**	**MER**	**LIZ**	**GEN**	**ERY**	**DPT**	**CRO**	**CIP**	**AMP**	**BC [µg mL^−1^]^1^**
1	**0.0625/1.1875**	0.0625	**1**	0.125	0.25	0.5	**2**	**2**	<0.25	16	**128**	4	0.25	0
<1	**0.0625/1.1875**	**0.0625**	**0.5**	**0.125**	**0.125**	0.125	**1**	**2**	<0.25	**16**	**128**	**4**	0.25	1
<1	**0.0625/1.1875**	**0.0625**	**0.25**	**0.0625**	0.0625	**0.0625**	**0.5**	**0.5**	<0.25	**16**	**64**	**1**	**0.125**	2
<1	**0.03125/0.59375**	<0.03125	<0.25	<0.0625	<0.0625	<0.0625	<0.5	<0.5	<0.25	<0.5	<1	<0.25	<0.0625	3
**Minimum Inhibitory Concentrations in µg mL^−1^**													**B**	
**VAN**	**T/S**	**TGC**	**TET**	**RIF**	**PEN**	**MER**	**LIZ**	**GEN**	**ERY**	**DPT**	**CRO**	**CIP**	**AMP**	**BC [µg mL^−1^] ^1^**
1	<0.03125/0.59375	0.0625	0.5	0.25	0.25	0.5	1	1	<0.25	16	64	4	0.25	0
<1	<0.03125/0.59375	<0.03125	<0.25	0.0625	0.0625	0.125	0.5	<0.5	<0.25	4	16	0.5	0.25	1
<1	<0.03125/0.59375	<0.03125	<0.25	<0.0625	0.0625	<0.0625	<0.5	<0.5	<0.25	<0.5	1	<0.25	<0.0625	2

^1^ Adaptation in the presence or absence of BC.

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
