# Peer review of "Benzalkonium Chloride Induces a VBNC State in Listeria monocytogenes"

_microorganisms, 2020, doi:10.3390/microorganisms8020184_

Round 1

Reviewer 1 Report

A very well, clearly written manuscript, that can be published as is. Just the column width of tables should be increased so that labels appear in full and not cut off.

The topic of this manuscript is of interest to food safety practitioners, as it proves that the highly infectious L. monocytogenes can build up some resistance to chemical cleaners provoking the VBNC state. Moreover, the fact that the culture methods would not be able to count the adapted L. monocytogenes cells gives a false sense of safety, which can have negative implications when assessing food processing and food production environments. There is therefore a need to investigate the suitability of other detection/counting methods for chemical resistant L. monocytogenes

Author Response

Thank you for your very positive feedback to our manuscript!

We have formatted the table accordingly to the author guidelines. As tables should not be rotated, we see no other option to present these data. If the manuscript will be accepted, we will discuss this table in the stage of proof reading. 

Reviewer 2 Report

It is interesting study. I think it is well designed.

However, please check:

L 145-146; 181-182; 195-196

Different letters above bars within panels indicate significant differences (p < 0.05) according to two-145 way analysis of variance (ANOVA).

There si no letters in Figures

L 259

The following are available online at www.mdpi.com/xxx/s1

The link is not valid

Author Response

Thank you very much for critical review!

Point1: L 145-146; 181-182; 195-196: According to your comments, we have added in the revised manuscript version the letters in figure 1, 2 and 3 to indicate significant differences.

Point 2: L 259: It is true that the online link to www.mdpi.com/xxx/s1 does not work. This link will probaly work after publication. However, supplementary figure S1 should be available elsewhere on the MDPI reviewing platform.